# Multicomponent Lipid Nanoparticles for RNA Transfection

**DOI:** 10.3390/pharmaceutics15041289

**Published:** 2023-04-20

**Authors:** Nataliya Gretskaya, Mikhail Akimov, Dmitry Andreev, Anton Zalygin, Ekaterina Belitskaya, Galina Zinchenko, Elena Fomina-Ageeva, Ilya Mikhalyov, Elena Vodovozova, Vladimir Bezuglov

**Affiliations:** 1Shemyakin-Ovchinnikov Institute of Bioorganic Chemistry, Russian Academy of Sciences, Moscow 117997, Russia; natalia.gretskaya@gmail.com (N.G.); akimovmike@gmail.com (M.A.); cycloheximide@yandex.ru (D.A.); belitskayakatya@yandex.ru (E.B.); zgn55@yandex.ru (G.Z.); evfa57@gmail.com (E.F.-A.); ilya.mikhalyov@gmail.com (I.M.); elvod@lipids.ibch.ru (E.V.); 2Department of Translational Medicine, National Research Nuclear University, Moscow Engineering Physics Institute, Moscow 115409, Russia

**Keywords:** lipid nanoparticles, cationic lipids, mRNA transfection, siRNA transfection, GM3 ganglioside

## Abstract

Despite the wide variety of available cationic lipid platforms for the delivery of nucleic acids into cells, the optimization of their composition has not lost its relevance. The purpose of this work was to develop multi-component cationic lipid nanoparticles (LNPs) with or without a hydrophobic core from natural lipids in order to evaluate the efficiency of LNPs with the widely used cationic lipoid DOTAP (1,2-dioleoyloxy-3-[trimethylammonium]-propane) and the previously unstudied oleoylcholine (Ol-Ch), as well as the ability of LNPs containing GM3 gangliosides to transfect cells with mRNA and siRNA. LNPs containing cationic lipids, phospholipids and cholesterol, and surfactants were prepared according to a three-stage procedure. The average size of the resulting LNPs was 176 nm (PDI 0.18). LNPs with DOTAP mesylate were more effective than those with Ol-Ch. Core LNPs demonstrated low transfection activity compared with bilayer LNPs. The type of phospholipid in LNPs was significant for the transfection of MDA-MB-231 and SW 620 cancer cells but not HEK 293T cells. LNPs with GM3 gangliosides were the most efficient for the delivery of mRNA to MDA-MB-231 cells and siRNA to SW620 cells. Thus, we developed a new lipid platform for the efficient delivery of RNA of various sizes to mammalian cells.

## 1. Introduction

Recently, gene therapy has become a developed area of medical technology, allowing one to cope with both genetically determined (congenital) and acquired diseases. The range of therapeutically significant nucleic acids is quite large, including plasmid DNA, mRNA, microRNA, small interfering RNA, and antisense oligonucleotides [1]. Both viral vectors [2] and non-viral platforms are used to deliver nucleic acids [1,3]. Viral vectors, like natural viruses, contain a nucleic acid in a complex with a cationic protein, which compensates for the negative charge of the phosphate groups of the nucleic acid. Non-viral platforms based on lipid [4] or mixed lipid–polymer particles [5] do not contain a protein component, and its role is played by a positively charged lipid.

Outbreaks of viral diseases, especially the SARS-CoV-2 virus pandemic, have significantly intensified work on the development of vaccines based on mRNA encoding viral proteins using lipid nanoparticles (LNPs) [6,7,8]. Lipid nanoparticles can be divided into two large groups depending on the type of lipid that binds the nucleic acid: type 1 LNPs with pH-sensitive ionizable lipids and type 2 LNPs containing cationic lipids. Type 1 LNPs usually use lipids with a tertiary amino group that acquires a positive charge after protonation, which allows for the binding of a nucleic acid, such as a capsid protein. Such LNPs can be considered virus-like particles, where the nucleic acid in a complex with the ionizable lipid is surrounded by an outer lipid membrane. The main method for their preparation is the use of microfluidic technology based on mixing a lipid solution in an organic solvent with an aqueous solution of nucleic acid at a certain pH [9].

LNPs of the second type contain lipids with a stable positive charge on the quaternary nitrogen atom [10]. These particles can be separately prepared and then complexed with a nucleic acid, although the use of microfluidic technology to simultaneously form a nucleic acid–LNP complex (lipoplex) is also applicable [11].

In addition to functional cationic or ionizable lipids, which are necessary for binding genetic material and fusion with the cell membrane, lipid nanoparticles include several so-called auxiliary lipids: structural lipids, such as distearoylphosphatidylcholine, dipalmitoylphosphatidylcholine to create a bilayer membrane, cholesterol to ensure membrane compaction, PEG-containing lipids that stabilize nanoparticles, and dioleoylphosphatidylethanolamine (DOPE) to destabilize endosome membranes and release nucleic acids. The role of these lipids in delivery systems is described in detail in the review of [12].

Ionizable type 1 LNPs are synthetically obtained, and their structures significantly differ from natural lipids. For example, in the world’s most used SARS-CoV-2 mRNA vaccines from Moderna and Pfizer/BioNTech, branched alkyl carboxylic acids (ALC-0315 Pfizer/BioNtech’s BNT162b lipid) or alkyl alcohols (Moderna’s SM-102 lipid) are used as ionizable lipids. The SM-102 structure includes an ionizable head group ester-linked to a hydrocarbon chain [13,14].

The most used synthetic cationic lipids in LNPs of the second type are simpler in structure, do not contain branches in the alkyl part, and mimic natural lipids to a greater extent. Thus, the cationic lipid DOTMA (1,2-di-O-octadecenyl-3-trimethylammonium propane) (chloride or bromide salt) used for DNA and mRNA delivery [15,16] is structurally close to natural dialkyl glycerols. Another popular cationic lipoid, DOTAP (1,2-dioleoyloxy-3-[trimethylammonium]-propane), is a trimethylamine analog of dioleoylglycerol [17] and is more biocompatible than the ionizable lipids mentioned above. The structures of synthetic cationic lipids are diverse in both the structure of the polar head and the structure of the hydrophobic part (for example, see [18]).

However, despite the great variety of synthetic cationic lipoids, the potential of natural cationic lipids is only exploited to a limited extent. As far as we know, there are no data on the use in LNPs of natural acylcholines, the acetylcholine analogs containing a saturated or unsaturated fatty acid residue. Such lipids contain only one quaternary ammonium group and are true cationic lipids [19]. Acylcholines, especially those containing polyene fatty acids, have a variety of biological activities [20,21] and could presumably confer additional functionality to cationic LNPs, such as the ability to interact with nicotinic receptors [20]. The acetylcholine receptor (AchR) can be used as a target for the delivery of therapeutically relevant nucleic acids. Thus, cationic and anionic liposomes containing siRNA were developed to inhibit the prion peptide in AchR-expressing cells. A synthetic peptide, RVG-9r, was used to target the particles to AchR [22]. The same AchR-targeting approach was used to deliver siRNA via the BBB [23]. It could be suggested that liposomes containing acylcholines can also be used for targeted delivery to AchR-expressing cells.

As a rule, LNPs of the second type, if they do not contain a hydrophobic polymer, are liposome-like particles with a bilayer membrane. On the other hand, if the internal part of each lipid particles is a hydrophobic core composed of polymer molecules (hybrid lipid–polymer nanoparticles) or natural lipids such as squalene and trimyristin [24], then the outer lipid membrane is a monolayer. It was important to compare the efficiency of the transfection of both types of particles (the lipid membrane of which contains the same types of lipids) under equal conditions since we could not find the results of such a comparison in the published data.

During the construction of LNPs with targeted properties or enhanced immune responses, we also turned our attention to particles containing GM3 gangliosides. The GM3 ganglioside, structurally the simplest monosialoglycosphingolipid carrying a terminal sialic acid residue on the lactosylceramide molecule, has a wide spectrum of biological activity [25,26], and its inclusion in LNPs can enhance the transport of particles into cells with a surface-exposed CD169 (Siglec-1) receptor. Decoration with GM3 gangliosides was used in the construction of artificial virus-like particles (gold nanoparticles wrapped in a membrane) [27,28] and particles built based on a polymer core and a lipid membrane [29].

The main objectives of this study were:1.To elucidate the possibility of using acylcholines as cationic lipids, either alone or in combination with DOTAP, for the formation of LNPs and lipoplexes with RNA.2.To compare the efficiency of the mRNA cell transfection of model LNPs with and without a core of non-polar lipids.3.To study the transfection activity of LNPs containing GM3 gangliosides.

We developed compositions of multicomponent lipid nanoparticles (LNPs) containing cationic and structural lipids in equal proportions and auxiliary surfactants that were capable of effectively delivering mRNA to mammalian cells. The replacement of the cationic lipid DOTAP with oleoylcholine destabilized the secondary structure of mRNA and led to a significant decrease in the transfection activity of LNP-based lipoplexes with oleoylcholine. Nanoparticles with a solid hydrophobic core based on cholesterol acetate and coconut oil triglycerides were more efficient in cell transfection compared with nanoparticles in which cholesterol acetate was replaced by squalene. However, core-bearing LNPs were less efficient in transfecting the HEK 293T cell line compared with non-nucleated particles. The incorporation of GM3 gangliosides into LNPs significantly increased the transfection ability of mRNA in these functionalized LNPs against SW620 colon tumor cells. The developed LNPs were effective in promoting cell transfection with both mRNA and siRNA.

## 2. Materials and Methods

### 2.1. Reagents

1,2-Dioleoylphosphatidylcholine (DOPC), 1,2-dioleoylphosphatidylethanolamine (DOPE), and DOTAP chloride (USP grade) were obtained from Lipoid GmbH (Heidelberg, Germany). Sorbitan monostearate (SPAN) was obtained from Sigma-Aldrich (Merk, Buchs, Switzerland), and polysorbate 80 (PS80) and squalene (Sq) were obtained from Acros Organics, and coconut oil triacylglycerols (COTs) were obtained from commercial coconut oil.

DMEM, L-glutamine, sodium pyruvate, non-essential amino acids, penicillin, streptomycin, amphotericin B, MTT, trypsin solution in EDTA, Earle’s solution, glucose, and Versene’s solution were purchased from PanEco, Moscow, Russia. DMSO and Triton X-100 were purchased from Sigma-Aldrich, St. Louis, MO, USA.

Fetal bovine serum was purchased from PanEco, Moscow, Russia.

#### 2.1.1. GM3 Ganglioside

GM3 gangliosides were isolated from the commercially available brains of cattle according to the method presented in [30]. After the isolation of the fraction of monosialogangliosides, the latter was separated into individual components with column chromatography on Silica gel 100 via elution with 65:25:2 or 65:25:4 (*v*:*v*:*v*) chloroform–methanol–water systems. Fractions containing GM3 gangliosides were collected as the first fraction of sialic lipids. The course of chromatography was monitored by TLC in a 60:40:9 (*v*:*v*:*v*) chloroform–methanol–15 mM CaCl_2_ system, detection with a resorcinol reagent, using GM3 gangliosides from Biomol GmbH (Hamburg, Germany) as a reference.

#### 2.1.2. Cationic Lipids

DOTAP iodide or mesylate was synthesized from 1,2-dihydroxy-3-aminopropane (Sigma-Aldrich, St. Louis, MO, USA) using oleoyl chloride according to the method of [31].

### 2.2. mRNA Preparation

The reporter mRNA encoding extracellular luciferase Nanoluc was prepared from a DNA template containing the T7 promoter and customized 5′ and 3′UTRs, encoding the nano luciferase sequence with a signal peptide and a poly-A (50) tail, 905 nucleotides in size. The in vitro transcription reaction was performed using the RiboMAX™ Large Scale RNA Production System (P1300, Promega, Madison, WI, USA) using a modified nucleotide composition for capping and incorporating N1 methyl pseudouridine. The transcription reaction medium contained the following concentrations of nucleotides: cap analog (3′OMe-m7G)5′ppp(5′)(2′oMe-A)pG (cat no. ON-205, Hongene Biotech, Union City, CA, USA), 30 mM; rGTP (Promega), 7.5 mM; rATP (Promega), 20 mM; rCTP (Promega), 20 mM; and N1-Methylpseudo-UTP (NU-890S, Jena Biosciences, Jena, Germany), 20 mM. T7 transcription was performed in a thermal cycler for 3 h followed by digestion with DNase RQ1 according to the manufacturer’s protocol. RNA was purified via LiCl precipitation and dissolved in RNase-free water, and then it was stored in a refrigerator at +4 °C.

The RNA fold was modeled using the ViennaRNA Package 2.0 program available from http://rna.tbi.univie.ac.at//cgi-bin/RNAWebSuite/RNAfold.cgi (accessed on 17 February 2023); see also [32]. The default energy minimization parameters were used according to the model presented by (Turner, 2004). Data are presented for minimum free energy prediction.

### 2.3. LNP Preparation

Lipid nanoparticles were obtained with a three-stage procedure similar to that described previously [33]. The lipid composition, based on the preparation of 3 mL of a solution with a 1.5 mg/mL concentration, was lyophilized in a 5 mL vial. Then, 3 mL of a 10 mM citrate buffer, pH of 6.6 and preheated to 70 °C, was added. Hydration and the formation of the initial emulsion were carried out in a Sapphire, 1.3/2 TTC ultrasonic bath with an operating frequency of 35 kHz at 70–75 °C for 11 min. Next, the hot emulsion was stirred using a PT 1200 dispenser (Kinematica AG, Luzern, Switzerland) equipped with a PT-DA 12/2ZMEC-E157 turbo mixer at a maximum speed of 12 m/s for 3 min.

A 600 to 800 µL aliquot of the resulting nanoemulsion was extruded through Whatman Nuclepore track-etched polycarbonate membranes (Cytiva, Marlborough, MA, USA) with a pore size of 100 nm on a mini-extruder (Avanti, Alabaster, AL, USA). The initial nanoemulsion and formed LNPs were stored in a refrigerator at a temperature of +4–8 °C.

The base composition (mole %) was: cationic lipid (DOTAP) (45), phospholipid (DOPE) (16), cholesterol (31), Polysorbate 80 (PS80) (3), and SPAN 60 (sorbitan monostearate) (5).

Samples of LNP complexes with mRNA (lipoplexes) were prepared by mixing a water solution of 100 ng of mRNA with the calculated amount of LNP solution to achieve the pre-determined ratio of cationic lipid–nucleic acid (N/P).

### 2.4. DLS Measurements

The size and surface charge of LNPs were assessed before and after complexing with mRNA. First, 50 μL aliquots of LNPs (prepared at 1.5 mg/mL) were diluted to 1 mL with a 6.6 pH citrate buffer, water, or saline and equilibrated at 23 °C in a Litesizer 500 (Anton Paar, Graz, Austria) cuvette holder before analysis. Particle size, polydispersity index (PDI), and zeta potential were calculated with the Kalliope™ software (Anton Paar, Graz, Austria). Each sample was analyzed for up to 100 runs or until acceptable stability was achieved in triplicate.

### 2.5. AFM Experiments

Atomic force microscope (AFM) and inverted microscope images were obtained on an SPM Nntegra base (NT-MDT) setup equipped with an Optem Zoom 125C inverted microscope (Qioptiq, Göttingen, Germany). The combination of the SPM setup with an inverted microscope made it possible to obtain AFM and light images from one area of the sample surface. Particle samples preliminarily diluted 20 times with water were applied to mica, rinsed with distillated water, and dried at normal pressure for a day. AFM images were obtained from scanning areas of 20 × 20, 16 × 16, and 4 × 4 µm (512 × 512 points) using an ETALON HAFM series cantilever (NT-MDT).

### 2.6. Electrophoresis

The electrophoretic separation of RNA and RNA-containing particles was carried out in agarose gel with a concentration of 1–2.5% based on a TBE buffer. Samples were applied to the gel using RNA Loading Dye (Thermo Fisher Scientific, Waltham, MA, USA). First, 10 µL of the sample (200 ng RNA) was mixed with 10 µL of the dye for application and applied to a 1.5% agarose gel containing 5 µL of ethidium bromide under a layer of a TBE electrophoresis buffer (89 mM Tris, 89 mM boric acid, and 2 mM EDTA). Samples with lipid particles were applied without using an application paint. Electrophoresis was performed at a constant voltage of 137 V. The results were visualized in ultraviolet light on a transilluminator. The length control of the products was performed based on compliance with the DNA markers in GeneRuler 1 kb Plus DNA Ladder (Thermo Fisher Scientific, Waltham, MA, USA).

### 2.7. Endoribonuclease Hydrolysis

To analyze the resistance of RNA complexes with lipid compositions to the action of RNases, 200 ng of RNA was incubated with 8 μL of LNPs prepared in water for 30 min at room temperature. As a control, 200 ng of RNA with 8 µL of water was used. After that, a 10,000× solution of SYBR Green I (Sigma-Aldrich) was added to each sample to a concentration of 1× in a Tris–borate–EDTA buffer (0.045 M Tris, 0.045 M H_3_BO_3_, and 1 mM EDTA at a pH of 8.0 without additional adjustment) in a volume of 0.8 μL and 0.05 units of RNase A (Thermo) in a volume of 1 µL of a 1/100 solution in water. The resulting mixture was vortexed, and fluorescence was analyzed in the SYBR Green I channel using a Gene4 apparatus (DNA Technology, Moscow, Russia) with an interval of 1 min for 30 min. An empty tube was used as a background; data are presented as signal-to-background ratios.

### 2.8. Cell Experiments

#### 2.8.1. Cell Culture

All cell lines were obtained from ATCC and cultured at 37 °C and 5% CO_2_.

Human embryonic kidney HEK 293T (ACTT CRL-3216), human breast cancer MDA-MB-231 (ATCC HTB-26), and human rectal cancer SW 620 (ATCC CCL-127) cells were cultured in a DMEM medium with a glucose content of 4.5 g/L supplemented with 4 mM of L-glutamine, 10% of fetal bovine serum, 100 U/mL of penicillin, 100 μg/mL of streptomycin, and 2.5 μg/mL of amphotericin B. Cells were passaged by treatment with a trypsin solution. The regular monitoring of mycoplasma contamination was carried out using a Mycoplasma Detection Kit (Jena Bioscience, Jena, Germany).

#### 2.8.2. Cell Viability Assay

Cell viability was assessed using an MTT test. To do this, the medium was removed and replaced with a 0.5 mg/mL MTT solution in Earl’s solution supplemented with 1 g/L of D-glucose and incubated for 1.5 h in a CO_2_ incubator. After that, an equal volume of 0.04 M HCl in isopropanol was added to the wells and incubated with stirring at 37 °C for 30 min. The stirring was performed using the plate incubator/shaker in the orbital rotation mode at a speed of 300 rev/min. The optical density of the solution was evaluated at wavelengths of 594 and 620 nm using a Hidex Sense Beta Plus microplate reader (Hidex, Turku, Finland). Each experiment was repeated at least three times. In some cases, a resazurin test was used to assess cell viability [34].

#### 2.8.3. mRNA Transfection and Luciferase Assay

To assess transfection efficiency, the cells were seeded at a density of 15,000/cm^2^ in 96-well plates in 100 µL of the cell culture medium the day before the addition of substances. Particle solutions in a volume of 4 μL were added to 2.5 μL of an RNA solution in water (40 ng/μL; calculation per well), mixed by pipetting, and incubated at room temperature for 30 min. After that, the whole mixture was added to the culture medium in the well, and the plate was stirred on a shaker for 5 s at a speed of 300 rpm.

To measure transfection, after 24, 48, or 72 h, 1 µL was taken from each well and placed into a black 384-well plate, after which 5 µL of a working solution for measuring luciferase activity (Promega) was added, mixed on a shaker for 5 s at a speed 300 rpm, and incubated in the dark for 5 min, after which the luminescence was measured using an IR Cutoff filter with a signal accumulation time of 10 s using a Hidex Sense Beta Plus microplate reader (Hidex, Turku, Finland).

#### 2.8.4. siRNA Transfection and PCR Analysis

Cells were seeded in a 96-well plate at a density of 15,000 cells in 100 μL of an appropriate complete nutrient medium per well the day before transfection.

Lipoplexes based on LNPs and anti-human-GPR55 siRNA (Thermo Fisher Scientific) were prepared by mixing LNPs and siRNA solutions at a rate of 4 µL of LNP solution in the base composition and 1 µL of the siRNA solution (1 pmol/µL) per well of a 96-well plate, as above, and added to the cells in a culture medium.

Knockdown efficiency was assessed 72 h later with RT-qPCR. Cells were cultured before RT-qPCR or substance treatment without changing the medium.

Total RNA was isolated using a Total RNA Purification Kit (Jena Biosciences, Jena, Germany) according to the manufacturer’s protocol. Residual genomic DNA was removed with DNase I (Thermo Fisher Scientific) according to the manufacturer’s protocol. For one RNA sample, 1 U of the enzyme was used. cDNA was synthesized using an MMLV reverse transcription kit (Evrogen, Moscow, Russia) with an oligo-dT primer. PCR was performed using an HS-Sybr Master Mix (Evrogen, Moscow, Russia), and the program was as follows: initial denaturation at 95 °C for a 3 min cycle, denaturation at 95 °C for 30 s, annealing at 57 °C for 30 s, DNA synthesis at 72 °C for 30 s for 35 cycles, and fluorescence detection at the end of each cycle. Amplification and detection were performed using a CFX-96 Touch apparatus (Bio-Rad Laboratories, Hercules, CA, USA). Primers were generated using the ITDNA PrimerQuest service (https://eu.idtdna.com/PrimerQuest (accessed on 7 February, 2023)) and validated using the NCBI Primer-BLAST service [35]. Their sequences were as follows: GPR55 direct 5′-ACTGATGTGCTTCCCTTTGAT-3′, reverse 5′-CCTGAACACTGGGTTATAAG-3′, POLR2A direct 5′-CCCAGCTCCGTT-GTACATAAA-3′, and reverse 5′-TCTAACAGCACAAGTGGAGAAC-3′. Reaction mixtures without reverse transcriptase and a cDNA template were used as negative controls.

### 2.9. Microscopy

Transmitted light microscopy was performed using an inverted Nikon Ti-S microscope equipped with an Andor Zyla 3.2 camera; the magnification was 100×.

Fluorescence was detected using the same microscope and a Semrock GFP-3035D filter cube. Image alignment and contrast correction were performed using the Nikon NIS Elements BR 4.0 software.

LNPs containing a fluorescent label were obtained using the base lipid composition (S1) with the addition of 0.6% tetramethyl-BODIPY-C3 acid (3-(4,4-difluoro-1,3,5,7-tetramethyl-4-bora-3*a*,4*a*-diaza-*s*-indacene-8-yl) propionic acid, as provided by Dr. I. A. Boldyrev, Institute of Bioorganic Chemistry, Russian Academy of Sciences). Lipoplexes with the labeled LNPs were prepared as described above.

### 2.10. Statistics

Each experiment was conducted in triplicate. The statistical processing of the results was carried out using the GraphPad Prism 9.3 software (GraphPad Software, San Diego, CA, USA). Data are presented as means ± standard errors. Data were compared using unpaired Student’s *t*-test for pairwise comparisons and ANOVA with Tukey’s post-test for multiple comparisons; *p*-values of 0.05 or less were considered significant.

## 3. Results

### 3.1. Preparation and Characterization of LNPs

Lipid nanoparticles without a hydrophobic core inside (LNPs) were formed from cationic and structural lipids in equal proportions and together comprised 92% of the composition. The main cationic lipid was DOTAP mesylate. Oleoylcholine (Ol-Ch) was also used as the sole cationic lipid or mixed with DOTAP. As structural lipids, a mixture of 30–31% of cholesterol (Ch) and 16% of phospholipid (DOPE, DOPC, or their mixture in a ratio of 1:1) was used (Table 1). During the formation of LNPs with a hydrophobic core (cLNPs), the amount of cationic lipids was proportionally reduced to compensate for the increase in lipid mass due to core components (14% and 7%). The hydrophobic core of core-containing lipid nanoparticles (cLNPs) consisted of equal parts of coconut oil triglycerides (COTs) and squalene (Sq) (similar to [33]) or cholesterol acetate (ChA), which mimics the cholesterol esters that are usually included in lipoprotein cores along with triacylglycerols. The surfactants of 5% stearoyl monosorbitan (SPAN60) and 3% polysorbate 80 (PS80) were used as additional LNP components in both variants, which provided additional stability to the nanoparticles in the solution. In addition, a functional lipid—the GM3 ganglioside (GM3)—was used in the LNPs (Table 1 and Figure 1).

The procedure for LNP and cLNP preparation comprised three stages. In the first stage, an anhydrous mixture of lipids was hydrated in an appropriate aqueous medium: in a citrate buffer (6.6 pH), in a citrate buffer (6.6 pH) with the addition of 0.9% NaCl, in a 0.9% NaCl solution, or in water. In all cases, the process was stimulated by external sonication and heating to 70–75 °C for 11 min. Next, the hot emulsion was intensively mixed with a dispersant at maximum speed for 3 min. In this stage, a stable suspension of lipid nanoparticles with a size of 200–250 nm was obtained; however, the standard deviation of the measured size could reach more than half of this value. For example, cLNPs containing a mixture of ChA and COTs in the core formed nanoparticles with a size of 218.4 ± 132.78 nm (PDI 0.23). After the final stage of extrusion through a membrane with a pore size of 100 nm, the size of these particles decreased and the suspension became more homogeneous (size of 158.7 ± 59.2 nm, PDI 0.18). A similar picture was observed for other compositions.

The sizes of particles of the basic composition prepared in the citrate buffer and the citrate buffer with the addition of NaCl to obtain an isotonic solution did not practically differ (205.3 ± 46.1 and 207.6 ± 48.2, respectively), though the particles prepared in water had a smaller size (147.4 ± 44.4). The size of the nanoparticles was practically independent of the lipid composition and averaged 176.7 nm (95% CI of 167.5 to 185.9). The mean polydispersity index (PDI) was 0.16 (95% CI of 0.15 to 0.18) (Table 2).

For the particles formed in the citrate buffer (6.6 pH), the zeta potential (or electrokinetic potential) regardless of lipid composition (except for LNPs with gangliosides) was approximately 4 mV. Only in the case of an equally equivalent mixture of DOPE/POPC structural phospholipids (S5) did the particles have a zeta potential close to zero (Table 2). LNPs prepared in water had a higher zeta potential than particles prepared in the citrate buffer (11.3 ± 0.4 versus 3.0 ± 1.0 for samples S1d and S1a and 12.4 ± 0.8 versus 42.5 ± 1.6 for samples S13a and S13d, respectively) (Table 2). The incorporation of GM3 gangliosides into the LNPs (samples S6–S8) resulted in a decrease in the zeta potential to approximately ±1 mV.

We prepared a model mRNA encoding the secretory Nanoluc luciferase protein and measured transfection effectiveness as a luminescence signal from the incubation medium upon the addition of a specific luciferin. The inclusion of mRNA into LNPs (sample S1) with the formation of a lipoplex led to an increase in particle size (from 205.3 ± 46.1 to 248 ± 55 nm) while maintaining the polydispersity index (PDI values of 0.12 and 0.13, respectively). The zeta potential of the lipoplex slightly decreased (3.0 ± 1.0 and 1.7 ± 0.8 mV, respectively). It should be noted that an excess of LNPs was used in the preparation of the lipoplex (see below), so the obtained values for both particle size and zeta potential refer to a mixed population of particles containing mRNA and original LNPs, which was demonstrated using atomic force microscopy (AFM).

AFM images of the LNPs (Figure 2a) showed that these particles were ellipsoidal. The diameter of most of them was about 100 nm; however, particles with a diameter of 150–200 nm were also present in the scanning field in smaller quantities. At the same time, it cannot be ruled out that the spread of particle sizes was much smaller since some particles could have been attached to the substrate in different configurations (as aggregates), which is reflected in the AFM images.

The lipoplex samples consisted of two types of particles: the original LNPs (Figure 2b) and particles with mRNA (Figure 2c). Figure 2c shows that the lipoplexes with mRNA were irregular in shape. The particle profile (representative object) had many local maxima, and the half-width was 180 nm. Since the half-height of the object (1.8 nm) did not exceed the bilayer thickness, the formation of a sandwich structure can be ruled out.

### 3.2. Completeness of mRNA Incorporation into Lipoplex and Protection of mRNA from Hydrolysis by Endonuclease

We controlled the completeness of mRNA inclusion using electrophoresis. There were no bands of free mRNA on the lipoplex electropherogram (Figure 3a). We also checked the absence of free mRNA in a solution of mRNA and LNP S1 lipoplexes prepared in water using the action of RNase A. As a rule, the formation of a complex of RNA with nanoparticles leads to its inaccessibility for the action of enzymes in solution if the incorporation of the nucleic acid is complete. On the contrary, if mRNA remains outside the complex with LNPs, it undergoes hydrolysis.

To study mRNA hydrolysis by RNase, we used the SYBR Green I (SG) fluorescence detection method. When SG is included in nucleic acid regions with paired nucleotides, the fluorescence of the dye increases by more than a thousand times [36]. The intensity of fluorescence can be used to judge the integrity of the nucleic acid since when it is hydrolyzed, the fluorescence intensity decreases. Under our experimental conditions, a solution of lipid particles with SG without mRNA did not exhibit fluorescence. During the incubation time (1 h and 40 min), the fluorescence of the standard mRNA sample did not change; during the same time in the solution with the added RNase A, the fluorescence dropped to the background value, which indicated the complete hydrolysis of the nucleic acid. In samples with lipoplexes and RNase A, we did not observe a drop in fluorescence. This fact indicates the complete protection of RNA from the action of the enzyme under experimental conditions and, accordingly, the complete inclusion of RNA in particles (Figure 3b).

During the formation of a lipoplex with most LNP compositions, the intensity of SG fluorescence did not change compared with the fluorescence of native mRNA (Figure 4). We only observed two exceptions, i.e., LNPs containing 1.7% of the GM3 ganglioside (sample S8) and LNPs with 44% of oleoylcholine from all cationic lipids (sample S13), for which we observed significant decreases in SG fluorescence intensity of 14% and 38%, respectively (Figure 4). At the same time, mRNA remained unavailable for hydrolysis by RNase. It can be assumed that the inclusion of gangliosides in the composition of LNPs and the use of a DOTAP/Ol-Ch mixture as cationic lipids led to a change in the secondary structure of mRNA after the formation of a lipoplex with a decrease in the number of paired nucleotides. We simulated the secondary structure of mRNA using the ViennaRNA Package 2.0 web service using the minimization of the free energy of the molecule. Our calculations showed that 58.6% of the total 908 nucleotides were paired (Appendix A). LNPs with oleoylcholine as the only cationic lipid (sample S3) had the greatest effect on the mRNA structure; in this case, the decrease in the SG fluorescence intensity compared with the control was 52%.

### 3.3. Influence of Lipoplex Formation Conditions on Transfection Efficiency

First, we determined the optimal conditions for the formation of lipoplexes during the interaction of mRNA with LNPs. Nanoparticles were used in the base composition (sample S1). The time of incubation of LNPs with mRNA had only a slight effect on the efficiency of the transfection of HEK 293T cells with lipoplexes, but the maximum effect was observed when incubated with mRNA for 40 min (Appendix A). Subsequently, this incubation time was used in all experiments. We noted no significant differences in transfection efficiency with lipoplexes prepared at 26 °C or 32 °C; however, lowering the preparation temperature to 8 °C significantly reduced the amount of synthesized luciferase (Appendix A).

### 3.4. Viability of the Transfected Cells

We inspected cell viability upon transfection with lipoplexes via both microscopy and measuring cell respiration with an MTT assay, which is routinely used to assess cell viability. To assess the penetration of LNPs and lipoplexes into HEK 293T cells, we used particles containing a lipid-soluble fluorescent probe (BODIPY-C3) included in LNPs in the base composition (sample S1Flu). Microscopic inspection showed that all cells were attached to the substrate, almost arranged in a monolayer, had the correct shape, and looked alive and intact; practically no floating or deformed cells were observed. It can be assumed that treatment with the composition did not have a significant effect on cell viability and morphology (Figure 5a,b). For both samples, the accumulation of fluorescence in cells was observed, though it was more pronounced for the S1Flu sample that did not contain mRNA (Figure 5a,b). It is possible that the RNA-free particles penetrated cells better due to their greater charge compared with lipoplexes, which led to a difference in fluorescence intensity at 24 h of incubation. However, we have no experimental confirmation of this hypothesis.

The determination of the viability of HEK263T cells after 24 h of incubation with lipoplexes prepared from LNPs of various compositions using the MTT test did not reveal any decreases (Figure 5c). For some samples, some stimulation of cellular respiration was observed. However, no significant correlation with the LNPs’ lipid composition was observed, and this stimulation effect was not specifically investigated. The mRNA-free LNPs were also non-toxic to cells (Appendix A).

### 3.5. Optimization of the Composition of Lipid Nanoparticles

#### 3.5.1. Effect of Cationic Lipid Counterion Type on Transfection Efficiency

The search for the optimal composition of LNPs showed that the counterion of the cationic lipid affected the efficiency of cell transfection. Using DOTAP as an example, we investigated the effect of anion type on the efficiency of the transfection of HEK 293T cells. To this end, we synthesized DOTAP iodide and mesylate and used commercially available DOTAP chloride. The highest yield of luciferase was observed when DOTAP mesylate was applied as a cationic lipid; meanwhile, DOTAP chloride was the least effective in delivering nucleic acids to cells, and DOTAP iodide demonstrated an intermediate transfection efficiency (Figure 6). Subsequently, most of the other experiments were performed using DOTAP mesylate.

#### 3.5.2. Type of Cationic Lipid in LNPs

Next, we explored the possibility of replacing the cationic lipid in LNP formulations. We prepared samples containing oleoylcholine instead of DOTAP (sample S3), as well as samples containing a mixture of DOTAP and oleoylcholine in various ratios: 11% (sample S15), 22% (sample S14) and 44% (sample S13) oleoylcholine of the total amount of cationic lipid. Lipoplexes with mRNA were formed, and the efficiency of transfection with these samples of HEK 293T cells was determined. It turned out that the replacement of the cationic lipid DOTAP with oleoylcholine significantly reduced the efficiency of transfection (Figure 7). Lipoplexes, in which the cationic lipid was a mixture of DOTAP and oleoylcholine in various ratios, also showed a low transfection efficiency, although it was higher compared with that of pure oleoylcholine (Figure 7). It can be assumed that the inclusion of a single-chain lipid in the bilayer disrupted the LNP structure, which affected the efficiency of transfection.

Thus, the most successful LNP variant turned out to be nanoparticles in which DOTAP mesylate was the only cationic lipid.

#### 3.5.3. N/P Ratio

An important parameter in the binding of LNPs to mRNA is the ratio of the cationic lipid LNPs in terms of the number of quaternary amino groups and phosphate groups of the nucleic acid, the so-called N/P ratio. We determined the optimal ratio of this parameter according to the efficiency of the transfection of HEK 293T cells with lipoplexes prepared from LNPs (sample S1) at various N/P ratios (Figure 8). It was found that transfection was most effective at an N/P ratio = 15.

#### 3.5.4. Effect of Structural Phospholipids on Transfection Efficiency

We also investigated the effect of structural lipid type on the transfection efficiency of various human cell lines. The HEK 293T, human breast cancer MDA-MB-231, and human colon cancer SW 620 cell lines were used for these experiments. The efficiency of transfection with lipoplexes in which DOPE and DOPC served as structural phospholipids was compared. For HEK 293T cells, the replacement of DOPE with DOPC did not affect the efficiency of transfection. Unexpectedly, the equivalent mixture of these lipids was ineffective (Figure 9a), and this composition was excluded from further experiments. In contrast to HEK 293T cells, the transfection of the MDA-MB-231 cell line with lipoplexes with DOPC instead of DOPE was significantly lower than in the case of DOPE (Figure 9b), while in the SW 620 cell line, the picture was reversed and lipoplexes with DOPC had a significant advantage in transfection (Figure 9c).

### 3.6. Storage Functional Stability of Nanoparticles

An important parameter is the preservation of the ability of an LNP to form a functional lipoplex after a sufficiently long storage period.

We studied the functional stability of LNPs with a sample of the basic composition (sample S1) when stored in the temperature range from +4 °C to +8 °C (which is usually used in the storage of preparations of this kind) for 6 months.

The stability of the LNPs was checked in terms of the efficiency of the transfection of HEK 293T cells at certain time intervals (Figure 10) for 6 months. The obtained results showed that the LNPs retained their effectiveness well during the entire storage period.

### 3.7. Lipid Nanoparticles with a Hydrophobic Core

We also compared the transfection efficiency of HEK 293T cells with lipoplexes prepared from cLNPs containing a hydrophobic core and particles without a core. Coconut oil triglycerides and squalene or cholesterol acetate were used as the hydrophobic cores. cLNPs containing cholesterol acetate and coconut oil triglycerides in the core were more efficient in transfection than particles with a core of squalene instead of cholesterol acetate (Figure 11a).

A comparison of the transfection efficiency of HEK 293T cells with lipoplexes prepared from LNPs (sample S1) and cLNPs (samples S11 and S12) showed that lipoplexes without a hydrophobic core were more effective for this cell line (Figure 11b).

### 3.8. Lipid Nanoparticles with Functional Lipid

In addition to cationic lipids, which ensure the binding of nucleic acids to a lipid carrier, LNPs can include so-called functional lipids. Such lipids can direct lipoplexes to a specific target or impart additional therapeutically significant functions to lipoplexes. We used the GM3 ganglioside as a functional lipid. With the inclusion of 0.5 and 1 mole percent ganglioside (samples S6 and S7), the efficiency of the transfection of HEK 293T cells with the corresponding lipoplexes of HEK 293T cells decreased compared with lipoplexes without GM3, and these samples were excluded from further experiments. However, with an increase in the content of GM3 to 1.7% (sample S8), such lipoplexes did not differ from the control in terms of transfection efficiency (Figure 12a). Lipoplexes with and without 1.7% GM3 behaved similarly on the MDA-MB-231 cell line (Figure 12b). However, for SW 620 cells, which are difficult to transfect, the use of lipoplexes with gangliosides allowed for a sharp increase in the yield of the protein product (Figure 12c).

### 3.9. Lipoplex Transfection of siRNA

The use of small interfering RNAs to block the activity of certain genes has recently become a common strategy for the treatment of diseases, especially cancer. We tested the efficiency of the delivery of siRNA against the GPR55 receptor, which plays an important role in the development of the tumor process, to MDA-MB-231 triple-negative breast cancer cells. To this end, commercially available siRNA and LNPs with 1.7% GM3 were used (sample S8). In the MDA-MB-231 cells, a pronounced expression of the GPR55 receptor gene was observed, but after treatment with lipoplexes and siRNA, the expression of this gene was completely suppressed (Figure 13).

Thus, the lipid nanoparticle constructs we developed can be effectively used for cell transfection with both mRNA and siRNA.

## 4. Discussion

In this work, we investigated two types of cationic lipid nanoparticles: liposomes (LNPs) and vesicles with a hydrophobic core (cLNPs). Both types of LNPs are used to deliver nucleic acids to cells. These particles are based on cationic lipids, most often DOTAP [17]. Both binary DOTAP–lipid systems, where phospholipid (DOPE) [37] or cholesterol [38] is used as a lipid, and various multicomponent systems [12,39,40] are used.

The composition of the LNPs we designed consisted of three main types of constituents: cationic lipids (a single lipid or a mixture of two), structural lipids represented by a mixture of cholesterol and phospholipid (one or two), and additional components (nonionic detergents). As the cationic lipid, either DOTAP or oleoylcholine (alone or mixed with DOTAP) was used. These lipids provided the main function of LNP–RNA binding. Despite the ability of DOTAP to independently form complexes with nucleic acids and form stable liposomes, the addition of natural lipids to LNPs increases the efficiency of transfection [41]. The substitution of oleic acid in the DOTAP molecule for saturated acids ranging from stearic to lauric acid makes liposomes from this cationic lipid unstable [41]. In these experiments, the authors synthesized DOTAP and its analogs as methanesulfonate (mesylate). In other works, the authors used DOTAP hydrochloride [42]. We compared the transfection efficiency of LNP lipoplexes prepared from the same components but using DOTAP with different counterions—chloride, iodide, and mesylate. It turned out that lipoplexes with DOTAP mesylate were the most effective transfectants, and the counterions could be ranked in the following order of transfection efficiency increase: chloride < iodide < mesylate.

To the best of our knowledge, acylcholines have not previously been used as a cationic lipid. In this work, we chose oleoylcholine (Ol-Ch) as the acylcholine in order to not change the fatty acid composition of lipids, the LNP components that only contain oleic acid (DOPE, DOPC, and DOTAP). Ol-Ch is a single-chain cationic lipid with a flexible hydrophobic chain. Most of the synthetic cationic lipids, such as DOTAP and its analogs [31,43] or ethylphosphatidylcholine [42], contain two hydrophobic chains, which ensures the stability of the bilayer membrane of cationic liposomes. Single-chain cationic lipids are used less frequently. Modified cholesterol (3*β*-[N-(N′,N′-dimethylaminoethane)-carbamoyl]cholesterol hydrochloride) [44] and synthetic cationic carotenoids [45] are of special interest. These substances, constructed via the chemical modification of natural lipids, contain a rigid side chain. Comparatively, Ol-Ch is a completely natural lipid that was detected in the serum of Bogalusa Heart Study participants [46]; it provides better LNP biocompatibility. Our experiments showed that Ol-Ch-based LNPs could bind mRNA, but the efficiency of transfection with such LNPs was significantly inferior to that of DOTAP–LNPs. Mixed LNPs containing both DOTAP and Ol-Ch were also less effective as transfection agents. At present, we cannot explain the reasons for this low efficiency of Ol-Ch in mixed LNPs. One possible explanation is the ability of Ol-Ch to influence the secondary structure of mRNA. Both Ol-Ch–LNPs and mixed DOTAP-Ol-Ch–LNPs during the formation of lipoplexes change the number of paired nucleotides in mRNA, thus changing its secondary structure, as shown by the intensity of SYBR Green I fluorescence (see below). Moreover, Ol-Ch–LNPs had the greatest effect, reducing the number of paired nucleotides by about 50% during the formation of lipoplexes with mRNA.

We used mixtures of a phospholipid, mostly DOPE, and cholesterol as structural lipids. DOPE can be considered an ionizable lipid, which facilitates the release of LNP contents into endosomes. At acidic pH values, DOPE undergoes a phase transition from a lamellar phase to an inverted hexagonal phase, which contributes to the destabilization of the endosomal membrane [47,48]. Unlike DOPE, whose molecule has a conical shape due to the small size of the polar head, the DOPC molecule is a cylinder and easily forms bilayer structures. LNP systems containing DOPC are more stable than LNPs with DOPE [49]. In our experiments, we noted no differences between DOPE–LNPs and DOPC–LNPs during mRNA transfection into HEK 293T cells. However, in the case of the MDA-MB-231 cell line, DOPC–LNP lipoplexes were an order of magnitude less efficient transfection agents, while on SW 620 cells, the transfection efficiency was reversed and DOPE–LNP lipoplexes were seven times less efficient.

The efficiency of the transfection of different cell lines depends on many factors. The most significant influence is exerted by the type of cells, the structure of their cytoplasmic membrane, the mechanisms of internalization of the transfecting agent, and the release of genetic material into the cytoplasm [50]. The importance of the process of the initial formation of granules of the cationic liposomes in a complex with DNA on the surface of the cell membrane as an initial factor that determines the efficiency of transfection was established in experiments with CHO cells. Helper lipids, DOPC and DOPE, play a decisive role in this process. For CHO cells, DOPC provides better interaction with the cytoplasmic membrane and more efficient endocytosis, which, according to the authors, is a rate-limiting step in the transfection of CHO cells [51]. The efficiency of transfection with lipoplexes also critically depends on the release of mRNA from endosomes, a process in which DOPE plays a decisive role due to its ability to destabilize the endosomal membrane [52]. We assume that for MDA-MB-231 breast cancer cells, the presence of DOPC in an LNP allowed for the better binding of the lipoplex to the cell membrane compared with SW 620 colorectal cancer cells; meanwhile, for the transfection of SW 620 cells, the endosomal escape process facilitated by the presence of DOPE seemed to be more significant.

The transfection efficiency of the DOPE and DOPC mixture-based lipoplex in a 1:1 ratio for the HEK 293T cells decreased by about three times compared with LNPs containing only one of these phospholipids. The transfection efficiency with a triple mixture of DOPC, DOPE, and DOTAP was noted by Pupo et al. for DNA-loaded liposomes [53]. These results highlight the importance of selecting the optimal structural lipids in the composition of cationic LNP lipids for each cell type.

The ratio of DOTAP/DOPE in most of the compositions we used, including the most effective (S1), was approximately 3:1, which correlates well with the data of Kim et al. on the effect of liposomal formulations with different weight ratios of DOTAP (as chloride) and DOPE: 1:0, 3:1, 1:1 and 1:3 [37]. It turned out that at a DOTAP/DOPE ratio of 3:1, the pcDNA-Luc transfection of three of the four cell cultures was the most efficient. Curiously, the authors of this paper noted that neither liposome size nor zeta potentials correlated with transfection efficiency.

Cholesterol is an important component of LNP structural lipids. It stabilizes LNPs’ lipid membranes [49] and is used in the preparation of LNPs for transfection [54,55] or drug delivery [56].

In addition to structural lipids, we used the auxiliary non-ionic surfactants SPAN60 and PS80 in the LNPs we constructed. SPAN80 containing oleic acid instead of stearic acid was used to prepare cationic liposomes for delivering siRNA to cells. A two-component 1:1 mixture of DOTAP and SPAN80 in combination with siRNA resulted in a decrease in the expression of the target gene by up to 60% [57]. In the LNP constructs, we used a significantly lower concentration of SPAN60 alone to increase the stability of the nanoparticles, especially in the case of compositions with a hydrophobic core. In addition, it has been found that three free hydroxyl groups of sorbitan residues contribute to the hydration of lipid surfaces and the capture of lipoplexes by cells [58]. PS80 was found to stabilize the nanoparticles and promote the solubilization of lipid components. In the preliminary experiments, we observed an incomplete transition to the aqueous phase of the initial thin lipid film in the stage of hydration during the preparation of LNPs if PS80 was absent in the composition. The branched structure of short units of polyethylene glycol forms a steric barrier on the surface of nanoparticles that prevents particle aggregation and promotes the sorption of nucleic acids on the surface of LNPs [59,60]. Bilayer particles based on nonionic surfactants (PS80 and others), called niosomes, are successfully used to deliver various drugs, including nucleic acids [61,62,63]. Niosomes generally contain equal amounts of PS80 and cationic lipids. We used the minimum amount of PS80 (3 mol %) to avoid undesirable effects of this surfactant on the signaling systems of body cells [64] when using the composition we developed in vivo.

The LNPs of most of the compositions had low zeta potential values. Perhaps this was due to the presence of PS80, which is a kind of oligomeric ethylene glycol attached to oleoyl sorbitan. It is known that the addition of PEGylated lipids to the composition of cationic liposomes reduces the value of the zeta potential. Thus, the addition of DPPE-PEG2000 to the DOTAP/DOPE composition was found to reduce the zeta potential from +42.0 ± 10 to almost zero (+3.2 ± 3) [65].

We determined the optimal conditions for the formation of lipoplexes during the incubation of LNPs and mRNA via the amount of synthesized luciferase. Neither an increase in the incubation time of more than 30 min or an increase in the incubation temperature to 32 °C significantly affected the efficiency of transfection. The optimal ratio of the positive charges of the cationic groups of lipids and the negative charges of mRNA phosphates (N/P) was determined as 15:1, which was also noted by other researchers [24].

The size of lipoplexes has a significant impact on the process of internalization and determines the efficiency of cell transfection [66,67]. There are rather conflicting data in the literature regarding the optimal size of lipoplexes for efficient transfection. In this study, we did not aim to create nanoparticles of various sizes. Most of the LNPs obtained by us had an average size of up to 200 nm, for which a clathrin-dependent mechanism of endocytosis is assumed [67]. Particle morphology determined with the structure of cationic or ionizable lipids also affects the efficiency of transfection, which has been shown for lipoplexes with DNA. Nucleic acids destabilize the lipid bilayer and lead to the formation of supramolecular lamellar (sandwich) or hexagonal structures [68].

Atomic force microscopy (AFM) has a high resolution and allows one to study the surface properties of liposomes [69]. Based on data from AFM microscopy, we observed a change in the LNP structure after the formation of a complex with mRNA. Instead of a smooth profile of an ellipsoidal particle, the lipoplex profile had multiple maxima, indicating the heterogeneity of the lipoplex surface. These heterogeneities may be associated with the branched mRNA structure located on the LNPs’ surfaces. The modeling of the secondary structure of mRNA showed that the nucleic acid had an extended structure with outgoing sections of paired nucleotides (Appendix A). The size of the lipoplex, measured on the AFM image (half-width of 180 nm), was slightly inferior to the size of the original LNPs reported based on DLS measurements (248 ± 55 nm). Based on this observation, it can be concluded that the lipoplex was formed through the complex formation of one lipid nanoparticle and one mRNA molecule. We confirmed the completeness of mRNA incorporation into the lipoplex using an electropherogram, which did not show the original mRNA band and any intermediate bands.

Previously, it was shown that to protect RNA from hydrolysis by RNases, there is no need to include nucleic acids inside lipid nanoparticles, as RNA sorption on LNPs’ surfaces is sufficient [4,33]. We also confirmed the resistance of mRNA in the lipoplex to the action of endonuclease RNase A. The absence of the hydrolytic cleavage of mRNA by RNase in lipoplex samples of various compositions allowed us to confirm the completeness of mRNA incorporation into LNPs.

In these experiments, we used SYBR Green I (SG) as an indicator, the fluorescence intensity of which increases by 3 orders of magnitude upon intercalation into double-stranded DNA [36]. Although this dye predominantly binds double-stranded nucleic acids, the ability of SG to be included in regions of paired nucleotides, accompanied by a significant increase in fluorescence intensity when it is almost completely absent in solution, makes it possible to use SG as an indicator of RNA integrity. The complexation of mRNA with LNPs protects the nucleic acid from RNase cleavage, and the initial signal from SG does not changes upon incubation with an enzyme. At the same time, free mRNA loses sections of paired nucleotides during hydrolysis, which leads to a decrease in the signal from SG. In our experiments, after 40 min of mRNA incubation with RNase, the signal from SG reached background values, which indicated the complete hydrolysis of the nucleic acid, as confirmed by the electrophoresis data. Enzymatic methods with SYBR Green I have been used for DNA analysis [70]. However, most publications have referred to the detection of double-stranded DNA and the use of SG for PCR analysis (see, for example, [71]), and we have not found any papers on SG application for direct RNA measurements.

The use of SG allowed us to additionally evaluate the nativeness of the secondary structure of mRNA during the formation of complexes with LNPs of various compositions. As in the case of the reaction with ribonuclease, with a decrease in the number of paired nucleotides, the signal from SG will become less intense and will not change if the mRNA structure does not undergo rearrangement with the loss of nucleotide conjugation. According to the data of modeling the secondary structure of mRNA using The Vienna RNA Web suite, 58.6% of the nucleotides in the structure of the mRNA we used were in a paired state. In samples in which the only cationic lipid was DOTAP, the SG fluorescence intensity did not differ from that of native mRNA. The binding of mRNA to LNPs with oleoylcholine as a cationic lipid led to a significant decrease in the number of paired nucleotides (the signal from SG, in this case, was no more than 48% of the control values). At the same time, nucleic acid hydrolysis by ribonuclease was not observed (Figure 4). We attribute the low efficiency of such lipoplexes during transfection to changes in the secondary structure of mRNA. It is possible that such lipoplexes inefficiently release mRNA in endosomes, which affects its availability for translation. With the simultaneous presence of DOTAP and Ol-Ch (44% of the total) in the composition of the LNPs, the decrease in the SG signal was less and amounted to 65% of the control values. At the same time, the efficiency of transfection with lipoplexes and LNPs of this composition was not much higher than when only using Ol-Ch as a cationic lipid. These results indicate that this method for detecting changes in the mRNA secondary structure only provides a preliminary basis for determining transfection efficiency. For example, a slight decrease in the number of paired nucleotides when the GM3 ganglioside (1.7%) was included in the LNP composition by 12% did not affect the efficiency of the transfection of HEK 293T cells with the corresponding lipoplex. However, this method allowed for a preliminary assessment of the nativeness of the secondary structure of mRNA during the formation of lipoplexes.

For the delivery of nucleic acids into cells, both bilayer liposomes with an aqueous inner content and monolayer liposomes containing hydrophobic core lipids are used [33,59,72]. We compared the transfection efficiency of both types of LNPs. The hydrophobic core was formed from lipids of two types; triacylglycerol and liquid squalene [33] or solid cholesterol acetate. Cholesterol acetate was previously used to prepare nanoparticles, both in a mixture with lecithin and surfactants [73] and as part of a hydrophobic core in a mixture with triolein [74]. A comparison of the transfection efficiency of nanoparticles with different compositions of the hydrophobic core revealed the advantage of cholesterol acetate over squalene. Direct comparison experiments between lipoplexes prepared from bilayer LNPs and monolayer cLNPs showed that for HEK 293T cells, bilayer LNPs are more efficient for transfection than monolayer cLNPs. However, it cannot be ruled out that in the case of larger mRNAs used in vaccine compositions, particles with a hydrophobic core may be preferable [33,59].

The functionalization of LNPs by introducing the GM3 ganglioside into the composition of lipids makes it possible to direct nucleic acids to antigen-presenting cells [29], which is important for the effectiveness of mRNA-based vaccines [75]. A comparison of the efficiency of the transfection of LNP cells of different lines containing GM3 gangliosides and basic nanoparticles showed that the inclusion of 1.7% gangliosides did not affect the efficiency of the transfection of HEK 293T and MDA-MB-231 cells. However, in the case of the SW 620 colon cancer cells, the advantage of functionalized LNPs was clear. It is known that the GM3 ganglioside receptor is CD169 (Siglec-1/Sialoadhesin), which allows for the targeting of liposomes with this ganglioside [76]. According to PCR data, the expression of the CD169 gene was found in all studied cells (Appendix A). It can be assumed that there are other non-receptor-mediated mechanisms that provide the better binding and subsequent endocytosis of lipoplexes with GM3 gangliosides in SW620 cells. It was found that the motility of SW620 cells is inhibited by exogenous GM3 gangliosides in the presence of transmembrane proteins CD9 [77] and CD82 [78], the expressions of which are upregulated in SW620 cells. The participation of these proteins in the binding of GM3 gangliosides cannot be ruled out, although the authors of these articles did not consider this aspect of ganglioside interaction with tumor cells.

The LNP compositions we developed make it possible to successfully deliver siRNA to cancer cells. RNA interference using small interfering RNA (siRNA) is widely used for research purposes to specifically knock out the genes of target proteins with a known mRNA sequence. This technology is also in demand for the fight against malignant tumors using a variety of lipid platforms for siRNA delivery to tumors [79,80]. In our experiments, using LNPs and siRNA against the GPR55 receptor gene, we managed to completely suppress the expression of this gene. The increased expression of the GPR55 receptor in cancerous tumors is an indicator of a poor prognosis of the disease outcome. Turning off the gene for this receptor can suppress the migration of cancer cells [81] and block the proliferative activity of the endogenous lysophosphatidylinositol ligand, which makes the GPR55 receptor a potential target for pharmacological intervention [82]. The ability of the developed LNPs to effectively deliver both small and larger RNAs into cells opens new prospects for multicomponent lipid platforms as promising pharmacological agents.

## 5. Conclusions

In this study, the formation of a model mRNA and LNP complex protected nucleic acids from the action of RNase, and the mRNA secondary structure did not substantially change (within the limitations of the detection method), which is important for the reliable transfection of mammalian cells.

It was found that the replacement of the cationic lipoid DOTAP with the natural Ol-Ch led to a sharp decrease in transfection efficiency, which may be associated with a disturbance of the secondary structure of mRNA that was previously shown for heterogeneous cationic liposomes [83] and a change in the intracellular fate of the lipoplex.

The nature of the counterion on the cationic lipid was important for transfection efficiency, and the best result was achieved with DOTAP mesylate.

A comparison of the effectiveness of hydrophobic core compositions composed of coconut oil triglycerides and squalene or cholesterol acetate showed the advantage of cholesterol acetate in the composition of the LNP core.

Core-bearing LNPs were less efficient in transfecting the HEK 293T cell line compared with non-nucleated particles.

For the HEK 293T cell line, both DOPE and DOPC were equally efficient as structural lipids in multicomponent LNPs. However, for the MDA-MB-231 cancer cell line, LNPs with DOPE were more efficient in transfection, while on SW620 cells, LNPs with DOPC had an advantage.

LNPs retained their potency to effectively transfect cells with mRNA for up to half a year when stored in a conventional refrigerator.

The incorporation of 1.7% GM3 gangliosides into LNPs significantly increased the transfection of SW620 cells with lipoplexes compared with non-functionalized LNPs. Ganglioside-functionalized LNPs also efficiently delivered siRNA to MDA-MB-231 cells.

The obtained data show the promise of the functionalization of cationic LNPs for their use as delivery vehicles for therapeutically important nucleic acids.

## Figures and Tables

**Figure 1 pharmaceutics-15-01289-f001:**
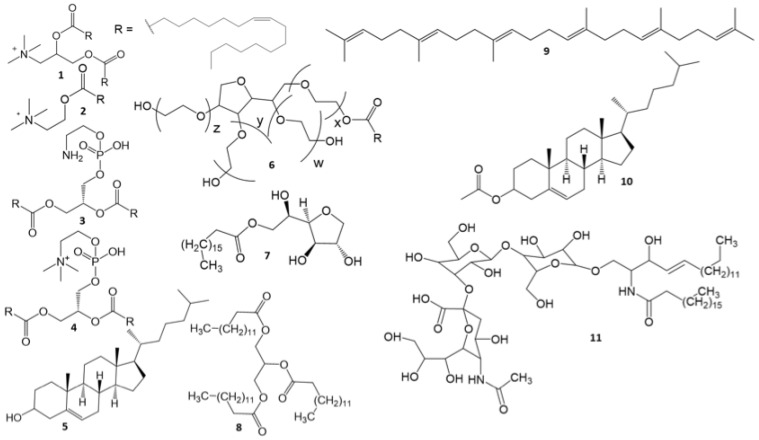
Structural formulas of the LNP and cLNP components. Cationic lipids: **1**, dioleoyltrimethylaminopropane (DOTAP); **2**, oleoylcholine (Ol-Ch). Structural lipids: **3**, dioleoylphosphatidylethanolamine (DOPE); **4**, dioleoylphosphatidylcholine (DOPC); **5**, cholesterol (Ch). Additional components: **6**, stearoylmonosorbitan (SPAN60); **7**, polyethylene glycol sorbitan monooleate (polysorbate 80 (PS80)), w + x + y = 20. Core lipids: **8**, coconut oil triglycerides (COTs); **9**, squalene (Sq); **10**, cholesterol acetate (ChA). Functional lipids: **11**, GM3 ganglioside (GM3).

**Figure 2 pharmaceutics-15-01289-f002:**
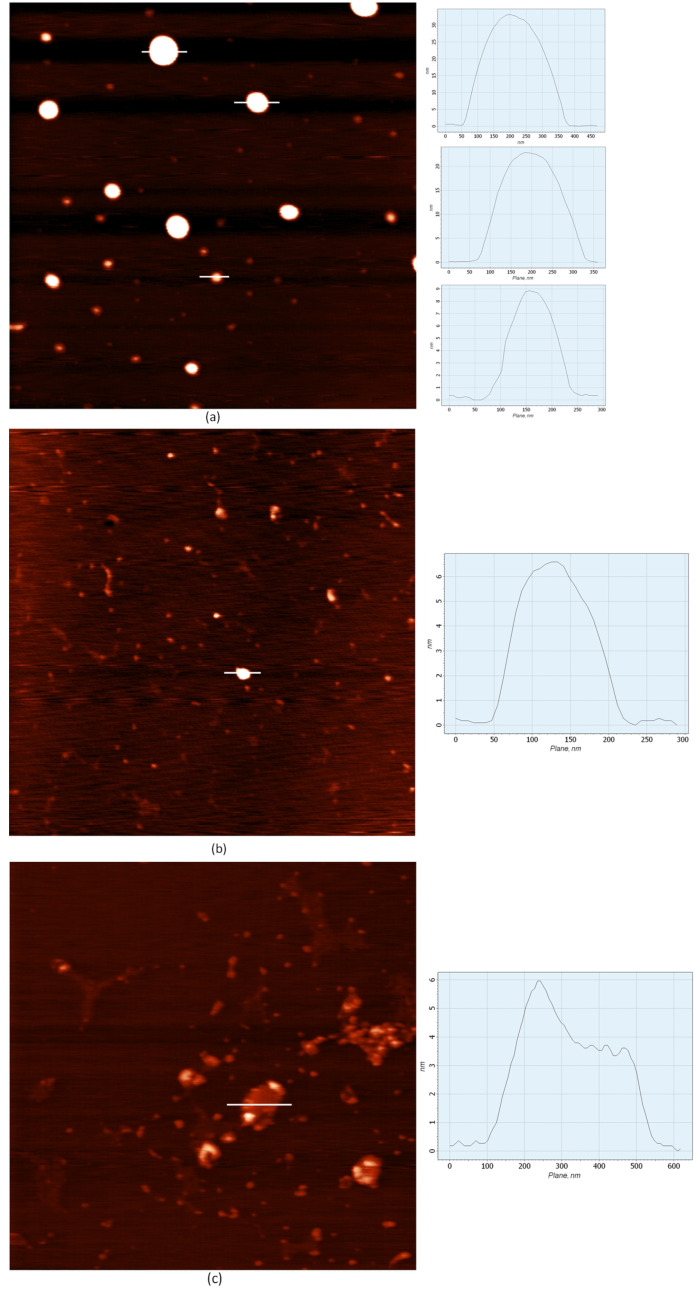
AFM images of LNPs and lipoplexes with mRNA formed in water. Experimental conditions: mi-ca, scanning area of 4 × 4 µm (512 × 512 pixels). (**a**) LNP sample S1 diluted to 1:20 with water; (**b**,**c**) LNP lipoplex (sample S1) and mRNA (N/P = 15) diluted to 1:20 with water. (**a**), the cross-sections from top to bottom, half-height: 16, 10.5 and 4.6 nm, half-width at half maximum: 115, 90 and 50 nm, (**b**), half-height 3.3 nm, half-width at half maximum 59 nm, (**c**), half-height 1.8, half-width at half maximum 180 nm. Representative samples from multiple scan fields are shown.

**Figure 3 pharmaceutics-15-01289-f003:**
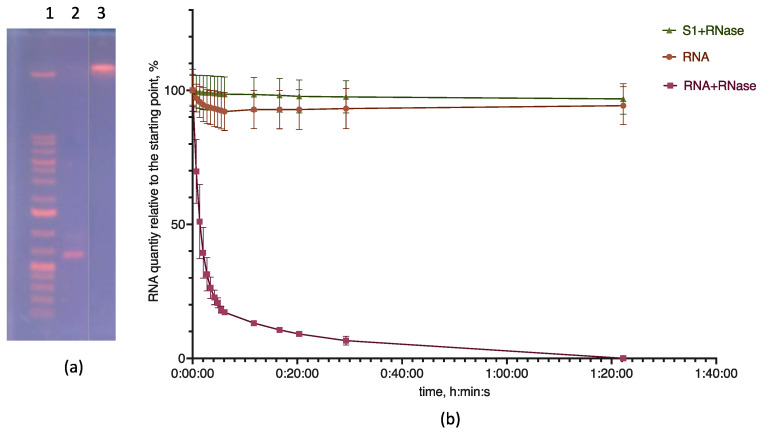
Electropherogram and kinetics of mRNA hydrolysis and mRNA–LNP lipoplex by RNase A. (**a**) Electropherogram of LNP (sample S1b) lipoplex sample: 1—markers; 2—mRNA; 3—LNP S1b and mRNA. (**b**) Kinetics of mRNA and lipoplex with LNP (sample S1b) hydrolysis. Values are given as percentages relative to the starting point of each sample. Tris–borate–EDTA buffer, 8.0 pH, 200 ng of mRNA, and 0.05 units of RNase A (Thermo). LNP sample S1 composition is given in Table 2, *n* = 3.

**Figure 4 pharmaceutics-15-01289-f004:**
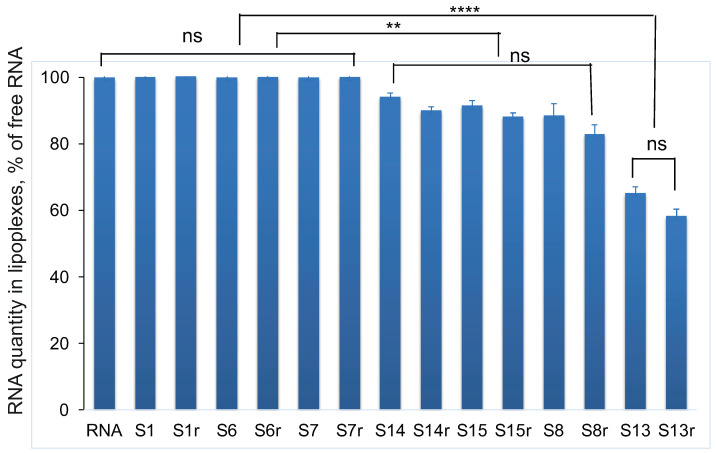
Quantity of RNA (relative to free RNA control, %) calculated from the average fluorescence intensity of SYBR Green I during the incubation of lipoplex samples for 30 min. Incubation of samples with RNase (index r) and without enzyme (no index); all samples were prepared in water. See Table 1 for LNP composition, mean ± SE. **, A statistically significant difference, *p* < 0.001; ****, a statistically significant difference, *p* < 0.0001; ns, not significant.

**Figure 5 pharmaceutics-15-01289-f005:**
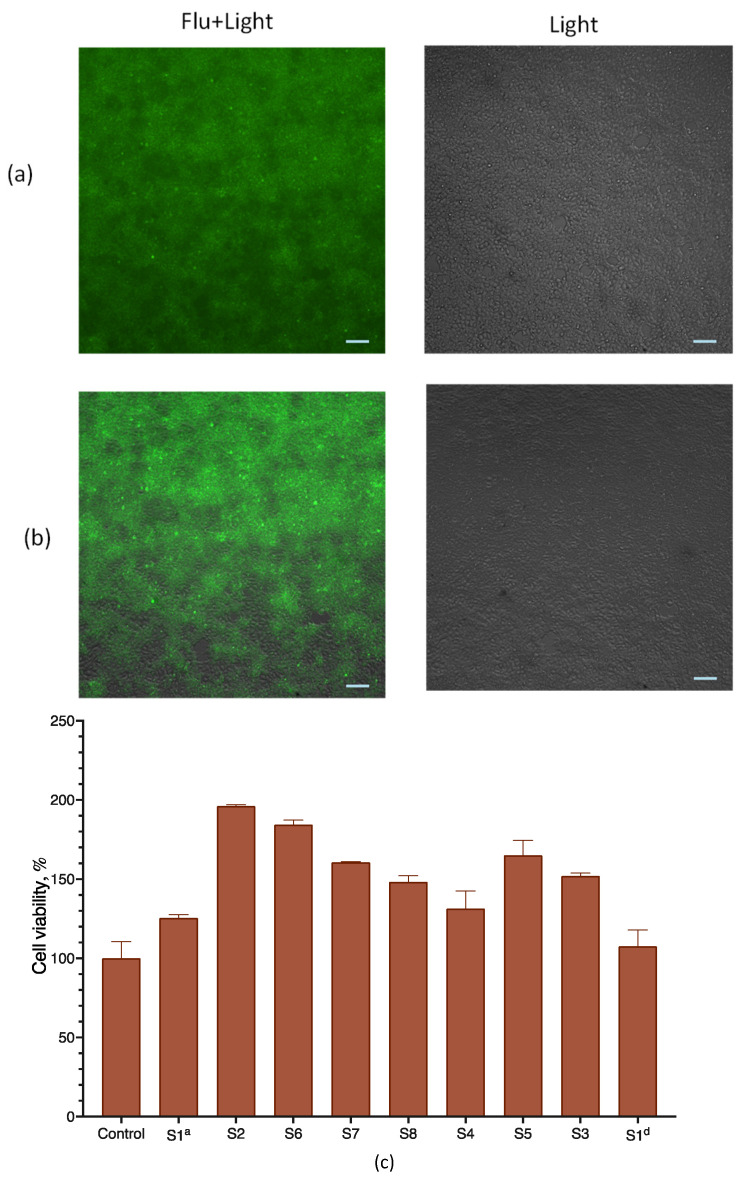
Viability assessment of HEK 293T cells. Microscopy of lipoplexes prepared from LNP (sample S1), labelled with BODIPY-C3 dye (S1Flu + mRNA, (**a**)) and BODIPY-C3 labelled LNPs (S1Flu, (**b**)); incubation time of 24 h and scale bar of 100 nm. Light, brightfield channel, Flu + Light, pseudocolor combined image, gray, brightfield channel; green, green fluorescence, (**c**) MTT test of HEK 293T cells after incubation with lipoplexes prepared from LNP samples for 24 h. See Table 1 for LNP composition; control—HEK 293T cells without particles, *n* = 3.

**Figure 6 pharmaceutics-15-01289-f006:**
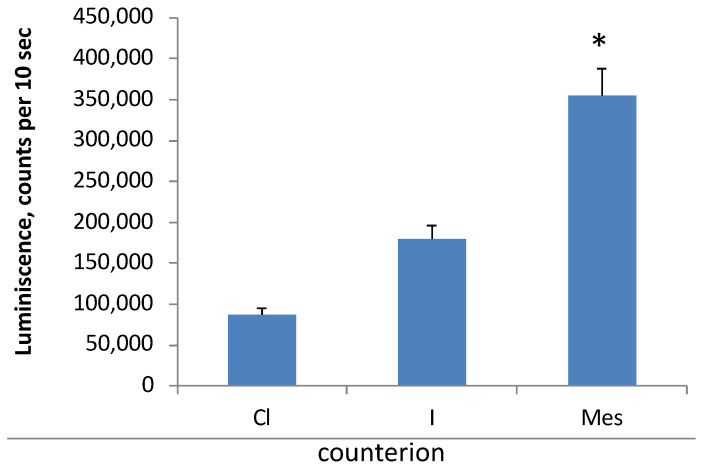
Transfection efficiency of HEK 293T cells with mRNA–LNP (sample S1) lipoplexes depending on various counterions of DOTAP: chloride (Cl), iodide (I) or mesylate (Mes). *, A statistically significant difference from samples with Cl and I counterions, *p* ≤ 0.05. Incubation time was 24 h, *n* = 3.

**Figure 7 pharmaceutics-15-01289-f007:**
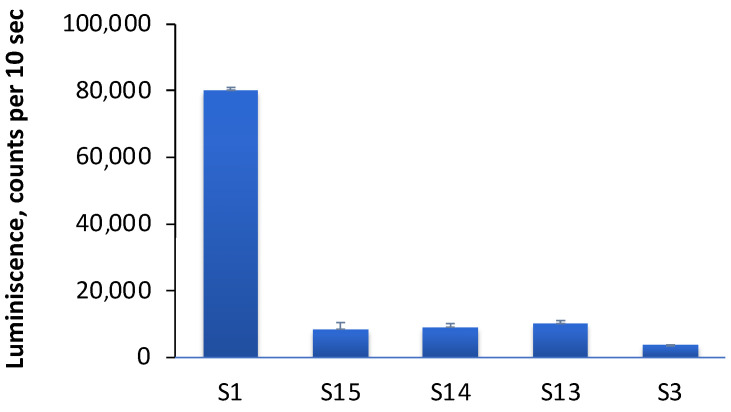
Transfection efficiency of HEK 293T cells with lipoplexes with various cationic lipids. Sample S1 contained 100% DOTAP, S3 contained 100% oleoylcholine, and samples S15–S13 contained a mixture of DOTAP and oleoylcholine in amounts of 11, 22, and 44%, respectively, of the total amount of cationic lipids (see LNP composition in Table 1). The untreated controls did not differ from the background signal (700 counts per 10 s). Incubation time was 24 h, *n* = 3.

**Figure 8 pharmaceutics-15-01289-f008:**
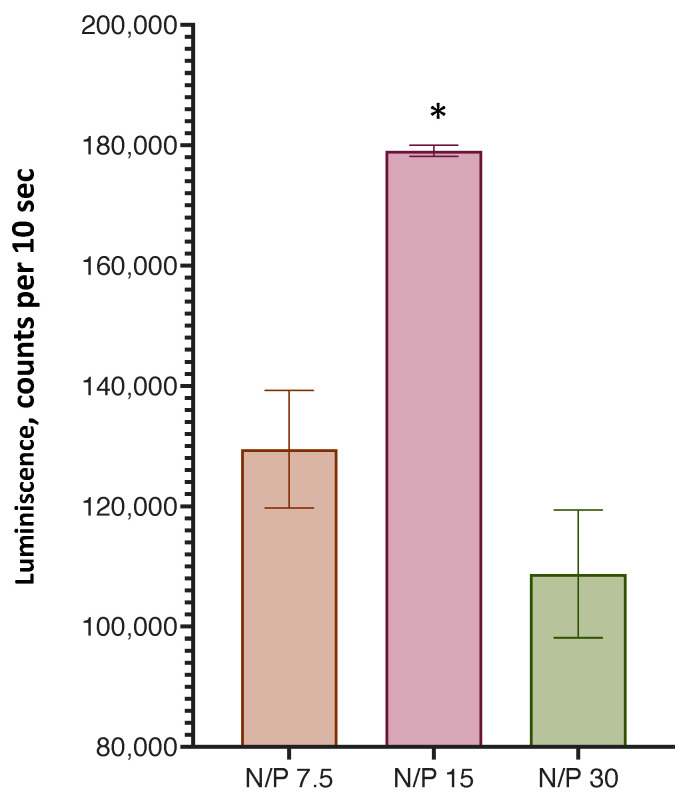
Transfection efficiency of HEK 293T cells with lipoplexes with different N/P ratios. *, A statistically significant difference from samples with N/P ratios of 7.5 and 30, *p* < 0.05. Incubation time was 24 h, *n* = 3.

**Figure 9 pharmaceutics-15-01289-f009:**
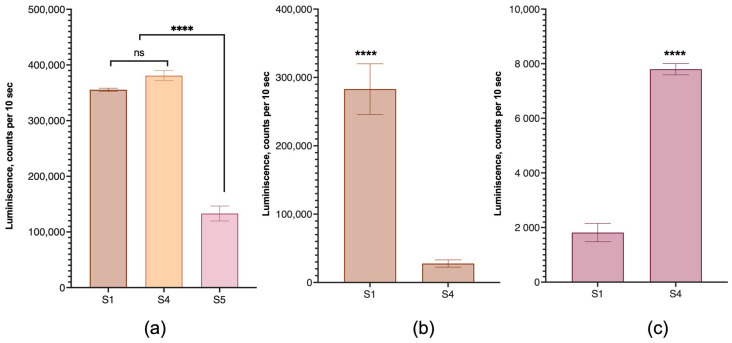
The efficiency of cell transfection with lipoplexes prepared from LNPs with different structural lipids. Phospholipid composition: 100% DOPE (sample S1), 100% DOPC (sample S4), and 50:50 DOPE/DOPC (sample S5). Incubation time was 24 h. (**a**) HEK 293T cells, (**b**) MDA-MB-231 cells, (**c**) SW 620 cells. The untreated controls did not differ from the background signal (700 counts per 10 s) and are not displayed. ****, A statistically significant difference, *p* < 0.0001, *n* = 3. ns, not significant.

**Figure 10 pharmaceutics-15-01289-f010:**
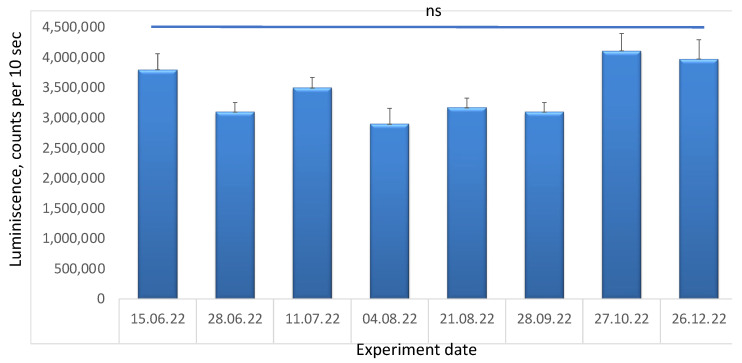
The efficiency of the transfection of HEK 293T cells with lipoplexes prepared from nanoparticles of the base composition (sample S1) stored at +4–8 °C. Incubation time was 24 h. ns. not significant, *n* = 3.

**Figure 11 pharmaceutics-15-01289-f011:**
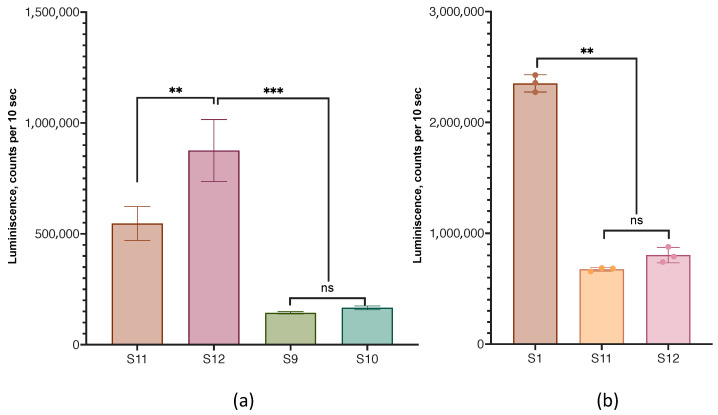
The efficiency of the transfection of HEK 293T cells with lipoplexes formed from LNPs with hydrophobic cores of various compositions. (**a**), Comparison of lipoplexes with different cLNP compositions. Samples S11 and S12 contained cholesterol acetate and coconut oil triglycerides in ratios of 7:7 and 4:4 mole %, respectively; samples S9 and S10 contained squalene and coconut oil triglycerides in ratios of 7:7 and 4:4 mole %, respectively (Table 1). (**b**) Comparison of the efficacy of lipoplexes prepared from LNPs (sample S1) and cLNPs (samples S11 and S12). The untreated controls did not differ from the background signal (700 counts per 10 s). Incubation time was 24 h. ns—not significant. ns. not significant, n = 3.

**Figure 12 pharmaceutics-15-01289-f012:**
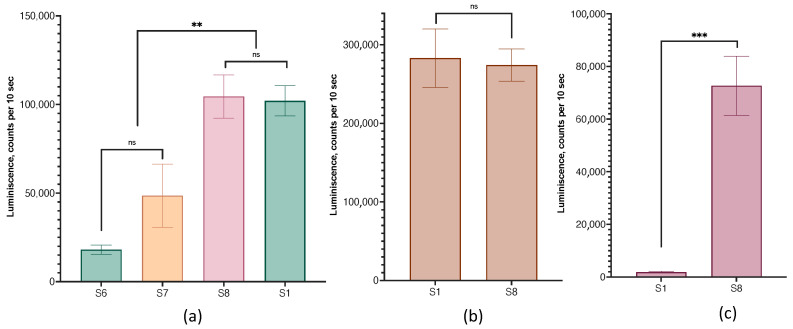
The efficiency of cell transfection with lipoplexes prepared from LNPs with GM3 gangliosides. Ganglioside content: 0.5% (sample S6), 1% (sample S7), and 1.7% (sample S8); control—sample S1 (Table 1). Incubation time was 24 h. (**a**) HEK 293T cells, (**b**) MDA-MB-231 cells, (**c**) SW 620 cells. ns—not significant. ns. not significant, *n* = 3.

**Figure 13 pharmaceutics-15-01289-f013:**
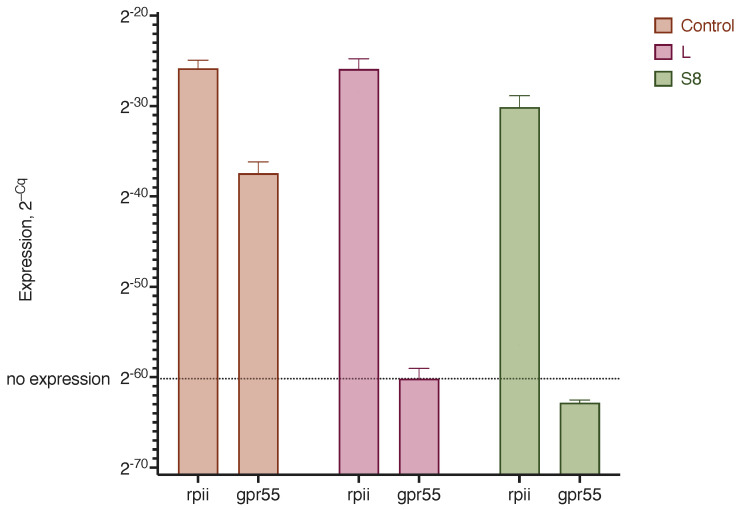
Cell transfection efficiency with lipoplexes prepared from LNPs with GM3 gangliosides and siRNA against the GPR55 receptor gene. Control, native MDA-MB-231 cells; LF, siRNA transfection with lipofectamine; S8, siRNA–LNPs S8 lipoplex transfection (1.7% GM3; see Table 1). Genes: rpii, POLR2A housekeeping gene; gpr55, GPR55 receptor gene. Incubation time was 72 h. RT-qPCR data and expression results are presented as values of 2 to the degree minus the number of the quantification cycle, *n* = 2.

**Table 1 pharmaceutics-15-01289-t001:** Compositions of lipid nanoparticles (mole percentage).

Sample No	Cationic Lipid * (%)	Phospholipid (%)	Core and Functional Lipid(s) (%)
S1 ^1^	DOTAP (46)	DOPE (16)	
S2 ^1^	DOTAP (Cl) (46)	DOPE (16)	
S3 ^1^	Ol-Ch (I) (46)	DOPE (16)	
S4 ^1^	DOTAP (46)	DOPC (16)	
S5 ^1^	DOTAP (46)	DOPE (8)/DOPC (8)	
S6 ^2^	DOTAP (45)	DOPE (16)	GM3 (0.5)
S7 ^1^	DOTAP (45)	DOPE (16)	GM3 (1)
S8 ^1^	DOTAP (44)	DOPE (16)	GM3 (1.7)
S9 ^3^	DOTAP (40)	DOPE (13)	Sq (7)/COT (7)
S10 ^4^	DOTAP (43)	DOPE (14)	Sq (3.5)/COT (3.5)
S11 ^3^	DOTAP (40)	DOPE (13)	ChA (7)/COT (7)
S12 ^4^	DOTAP (43)	DOPE (14)	ChA (3.5)/COT (3.5)
S13 ^1^	DOTAP (25.8)/ Ol-Ch (20.2)	DOPE (16)	
S14 ^1^	DOTAP (35.9)/ Ol-Ch (10.1)	DOPE (16)	
S15 ^1^	DOTAP (40.9)/ Ol-Ch (5.1)	DOPE (16)	

* Unless otherwise specified, the counterion of the cationic lipid is mesylate, in other cases, chloride (Cl) or iodide (I). ^1^ Cholesterol (30%); additional surfactants: SPAN60 (5%) and PS80 (3%). ^2^ Cholesterol (31%); additional surfactants: SPAN60 (5%) and PS80 (3%). ^3^ Cholesterol (26%); additional surfactants: SPAN60 (5%) and PS80 (3%). ^4^ Cholesterol (28%); additional surfactants: SPAN60 (5%) and PS80 (3%).

**Table 2 pharmaceutics-15-01289-t002:** Characteristics of lipid nanoparticles.

Sample No	Size, nm *	Hydrodynamic Diameter, nm	PDI *	Zeta Potential, mV
S1 ^a^	205.3 ± 46.1	191.4	0.12 ± 0.015	3.0 ± 1.0
S1 ^b^	207.6 ± 48.2	213.3	0.09 ± 0.011	3.3 ± 1
S1 ^c^	184.0 ± 39.4	193.3	0.1 ± 0.009	3.0 ± 0.8
S1 ^d^	147.4 ± 44.4	143.9	0.19 ± 0.012	11.3 ± 0.4
S2 ^a^	195.0 ± 74.9	181.3	0.23 ± 0.018	2.0 ± 0.6
S3 ^a^	166.8 ± 53.2	169.4	0.14 ± 0.009	4.3 ± 0.6
S4 ^a^	140.5 ± 44.3	138.2	0.17 ± 0.014	4.0 ± 0.8
S5 ^a^	159.3 ± 61.8	166.1	0.18 ± 0.024	0.4 ± 1
S6 ^a^	168.0 ± 51.8	172.4	0.17 ± 0.01	−0.8 ± 0.5
S7 ^a^	177.8 ± 56.2	181.5	0.15 ± 0.026	1.2 ± 0.6
S8 ^a^	177.8 ± 55.1	180.0	0.16 ± 0.007	0.7 ± 0.8
S9 ^a^	148.8 ± 71.1	154.0	0.19 ± 0.014	2.1 ± 0.7
S10 ^a^	143.4 ± 56.8	147.7	0.22 ± 0.014	0.2 ± 1
S11 ^a^	162.83 ± 59.2	158.7	0.18 ± 0.014	4.0 ± 0.4
S12 ^a^	171.5 ± 57.2	166.3	0.17 ± 0.017	3.6 ± 0.6
S13 ^a^	160.9 ± 60.7	152.2	0.20 ± 0.006	12.4 ± 0.8
S13 ^d^	118.3 ± 38.7	114.1	0.15 ± 0.024	42.5 ± 1.6
S14 ^a^	196.0 ± 89.8	169.4	0.19 ± 0.006	10.9 ± 0.6
S15 ^a^	190.7 ± 71.3	177.2	0.18 ± 0.005	10.4 ± 0.8

* Mean values ± SD are presented, n = 3. ^a^ Citrate buffer, 6.6 pH. ^b^ Citrate buffer, 6.6 pH, and 0.9% NaCl. ^c^ 0.9% NaCl. ^d^ H_2_O.

## Data Availability

The data can be made available upon request.

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
