# Peer review of "Multicomponent Lipid Nanoparticles for RNA Transfection"

_pharmaceutics, 2023, doi:10.3390/pharmaceutics15041289_

Round 1

Reviewer 1 Report

In this study, the authors develop multi-component cationic lipid nanoparticles (LNPs) with or without a hydrophobic core from natural lipids, to evaluate the efficiency of LNPs with the widely used ones and the ability of LNPs containing GM3 ganglioside to transfect cells with mRNA and siRNA. This is an interesting work, but requires major revision before further consideration.

1. Why there are only two repeats in many presented data?

2. The scale bar in Figure 2 and 6 should be clearly displayed.

3. The transfection should provided with pictures instead of only histograms.

4. The histograms throughout the whole manuscript should be unified with higher resolution and same SD expression ways.

5. The language can be improved and also there are many mistakes.

Reviewer 2 Report

see attached document

Reviewer 3 Report

Dear Authors,

The study entitled “Multicomponent lipid nanoparticles for RNA transfection” by Gretskaya et al. reports few lipid formulations designed as a transfecting tool. The manuscript contains significant amount of experimental data properly designed to explore the developed system’s properties and efficacy. While this reviewer believes the manuscript contains enough data to be published in Pharmaceutics some of the concerns listed below need to be answered/discussed by the authors to better justify the data and thus, make it clearer for the readers prior to the publication:

Additional explanation for Figure 7 would be appreciated. Abbreviations: I, Cl and Mes need to be elaborated in the caption or on the graph. In the section 3.5.1, the counterion I was not mentioned. 

In figure 10 sample S5 results are missing for MDA-MB-231 and SW 620 cell lines. Why was sample S5 tested only on HEK293T cells? Proper explanation needs to be provided.

Similarly in Figure 13 the results for samples S6 and S7 were missing for MDA-MB-231 and SW 620 cell lines.

Overall, the listed concerns do not decrease the scientific merit of the submitted manuscript. Thus, this reviewer believes the manuscript will be suitable to publish after minor changes.

Sincerely yours.

Round 2

Reviewer 1 Report

Accept in its current form.

Reviewer 2 Report

The authors addressed all mentioned issues in an appropriate matter.

Best regards